# Mutations in Filamin C Associated with Both Alleles Do Not Affect the Functioning of Mice Cardiac Muscles

**DOI:** 10.3390/ijms26041409

**Published:** 2025-02-07

**Authors:** Leonid A. Ilchuk, Ksenia K. Kochegarova, Iuliia P. Baikova, Polina D. Safonova, Alexandra V. Bruter, Marina V. Kubekina, Yulia D. Okulova, Tatiana E. Minkovskaya, Nadezhda A. Kuznetsova, Daria M. Dolmatova, Anna Yu. Ryabinina, Andrey A. Mozhaev, Vsevolod V. Belousov, Boris P. Ershov, Peter S. Timashev, Maxim A. Filatov, Yulia Yu. Silaeva

**Affiliations:** 1Center for Precision Genome Editing and Genetic Technologies for Biomedicine, Institute of Gene Biology, Russian Academy of Sciences, 119334 Moscow, Russia; lechuk12@gmail.com (L.A.I.); baykjulia@gmail.com (I.P.B.); aleabruter@gmail.com (A.V.B.); kubekina@genebiology.ru (M.V.K.); ul.okulova@gmail.com (Y.D.O.); inquisitive.lizard@gmail.com (T.E.M.); amanerge@gmail.com (N.A.K.); sv.daria.m@gmail.com (D.M.D.); yulya.silaeva@gmail.com (Y.Y.S.); 2Core Facility Center, Institute of Gene Biology, Russian Academy of Sciences, 119334 Moscow, Russia; pdsafonova@gmail.com; 3Institute for Regenerative Medicine, Sechenov University, 119991 Moscow, Russia; ksushakochegarova@gmail.com (K.K.K.); ershov_b_p@staff.sechenov.ru (B.P.E.); timashev_p_s@staff.sechenov.ru (P.S.T.); 4Laboratory of Molecular Oncobiology, Institute of Gene Biology, Russian Academy of Sciences, 119334 Moscow, Russia; 5V.A. Frolov Department of General Pathology and Pathological Physiology, Institute of Medicine, Peoples’ Friendship University of Russia (RUDN University), 6 Miklukho-Maklaya Street, 117198 Moscow, Russia; ryabinina.ayu@gmail.com; 6Laboratory of Molecular Technologies, Shemyakin-Ovchinnikov Institute of Bioorganic Chemistry, Russian Academy of Sciences, 117997 Moscow, Russia; a.a.mozhaev@gmail.com (A.A.M.); belousov@fccps.ru (V.V.B.); 7Group of Genome Editing Techniques, Center for Precision Genome Editing and Genetic Technologies for Biomedicine, Pirogov Russian National Research Medical University, 117513 Moscow, Russia; 8Federal Center of Brain Research and Neurotechnologies, Federal Medical Biological Agency, 117513 Moscow, Russia; 9Chemistry Department, Lomonosov Moscow State University, 119991 Moscow, Russia

**Keywords:** filamin C, flnc, cardiomyopathy, animal model

## Abstract

Filamin C (FLNC) is a structural protein of muscle fibers. Mutations in the *FLNC* gene are known to cause myopathies and cardiomyopathies in humans. Here we report the generation by a CRISPR/Cas9 editing system injected into zygote pronuclei of two mouse strains carrying filamin C mutations—one of them (AGA) has a deletion of three nucleotides at position c.7418_7420, causing E>>D substitution and N deletion at positions 2472 and 2473, respectively. The other strain carries a deletion of GA nucleotides at position c.7419_7420, leading to a frameshift and a premature stop codon. Homozygous animals (*Flnc*^AGA/AGA^ and *Flnc*^GA/GA^) were embryonically lethal. We determined that Flnc^GA/GA^ embryos died prior to the E12.5 stage and illustrated delayed development after the E9.5 stage. We performed histological analysis of heart tissue and skeletal muscles of heterozygous strains carrying mutations in different combinations (*Flnc*^GA/wt^, *Flnc*^AGA/wt^, and *Flnc*^GA/AGA^). By performing physiological tests (grip strength and endurance tests), we have shown that heterozygous animals of both strains (*Flnc*^GA/wt^, *Flnc*^AGA/wt^) are functionally indistinguishable from wild-type animals. Interestingly, compound heterozygous mice (*Flnc*^GA/AGA^) are viable, develop normally, reach puberty and it was verified by ECG and Eco-CG that their cardiac muscle is functionally normal. Intriguingly, *Flnc*^GA/AGA^ mice demonstrated better results in the grip strength physiological test in comparison to WT animals. We also propose a structural model that explains the complementary interaction of two mutant variants of filamin C.

## 1. Introduction

Filamin C (FLNC, γ-filamin) is an actin-binding protein, a member of the filamin family. FLNC localizes to the sarcolemma, Z-discs (in the case of striated muscle tissue), and intercalated discs (in the case of cardiomyocytes) [1,2]. In the normal dimerized state it interacts with numerous FLNC-binding proteins, including those responsible for normal sarcomere structure organization, signaling cascade members and cell junction proteins [3,4]. The main proteins that co-interact with FLNC are the Z-disc molecules such as myotilin, myozenin, myopodin, and calsarcins [5]. Moreover, it interacts with signaling molecules [6] and sarcolemma-associated proteins such as integrin β1 and sarcoglycan delta [7,8,9]. Furthermore, FLNC interacts with aciculin and XIRP (Xin active-binding repeat-containing proteins); these interactions are required to fulfill a function in muscle maintenance [10].

Similarly to other filamins, FLNC is a structural protein that has an actin-binding domain (ABD) made up of two calponin homology domains (CH), followed by 24 immunoglobulin-like (Ig-like) domains divided into ROD1 and ROD2 regions, and a C-terminal dimerization domain [11]. The ROD2 domain contains the majority of binding sites for FLNC-binding partners and is required for mechanosensing and muscle function support [5].

A list of mutations in the *FLNC* gene (https://www.omim.org/entry/102565; accessed on 13 November 2024) have been shown to be associated with pathology of various degrees of severity [12,13,14,15]. For example, heterozygous truncations of *FLNC* cause dilated cardiomyopathy and arrhythmia [5]. Aberrant protein folding and dimerization failure lead to aggregate formation, negatively affecting muscle function, which is characteristic of hypertrophic myopathy and myofibrillar myopathy. ROD2 domain mutations are often associated with hypertrophic cardiomyopathy [5]. 

Myopathy is a chronic progressive neuromuscular disorder characterized by primary muscle lesions. An example of myopathies is cardiomyopathy (CM)—any structural or functional pathology of cardiac muscle not related to a particular condition, such as ischemic heart disease, congenital heart defect or valve anomaly [16]. The major types of cardiomyopathies are dilated, hypertrophic and restrictive, all characterized by different pathological changes. The most common CM type is hypertrophic cardiomyopathy; it is diagnosed in 1 in 500 adults and is one of the primary causes of death [17].

Different mutations in the *FLNC* gene lead to various manifestations. Truncated FLNC variants (premature stop codon formation) can lead to DCM (dilated cardiomyopathy) [5]. Abnormal FLNC variants have 1–4.5% of patients who suffer from DCM [14,18,19,20]. DCM is characterized by the presence of contractile dysfunction and left ventricular dilatation, although abnormal loading conditions and severe coronary artery disease are not present [21,22]. Missense variants are typically associated with HCM (hypertrophic cardiomyopathy); the prevalence of such mutations in HCM cohorts is about 1.3–8.7% [18,23,24,25]. HCM is characterized by the presence of increased left ventricular wall thickness that cannot be solely explained by abnormal loading conditions [21,22].

Previously published work performed on mouse models has reported that deletion of exons 41–48 in the *Flnc* gene causes neonatal lethality in homozygotes due to a breathing disorder, while heterozygous animals had lower muscle weight and a reduced count of primary muscle fibers [26]. Other studies have shown that deletion of 16 C-terminal amino acid residues in the FLNC protein prevents the normal dimerization process, leading to muscle weakness and myofiber instability in heterozygous mice [27,28].

In this work we have generated two genetically edited mouse strains that carry distinct mutations in the ROD2 domain-coding region of the *Flnc* gene. We have performed morphological and microscopic analyses of *Flnc*-mutated mouse embryos, phenotype assessment of adult animals with filamin C mutations, including histological analysis of cardiac muscle and comparison of muscle performance. We have also conducted bioinformatics analysis of possible structural changes in mutant protein forms. Interestingly, compound heterozygous animals (which carry different mutant alleles of the *Flnc* gene) successfully develop and do not exhibit any cardiomyopathy signs, neither on the histology level nor in ECG (electrocardiography) and functional diagnostic assays.

## 2. Results

### 2.1. Two Mutant Mouse Strains Were Generated

From 460 transplanted embryos injected with the gene editing system, 36 F0 generation pups were obtained, of which 2 had the desired genetic modifications. Two animal strains were used in subsequent experiments—one of them carried a deletion of three nucleotides in exon 45 at position NM_001081185.2:c.7418_7420, and, thus, an amino acid substitution p.E2472D and Asn2473 deletion, and in the other, a GA deletion at NM_001081185.2:c.7419_7420 led to a frameshift mutation and a premature stop codon. The mutant alleles were designated *Flnc*^AGA^ and *Flnc*^GA^, respectively. Schematic representation of the mutations’ positions and Sanger sequencing results of F1 animals are shown in Figure 1.

To assess the viability of hetero- and homozygous carriers of novel mutations, monohybrid crossings of heterozygous animals of both mutant strains were set up: *Flnc*^wt/GA^ × *Flnc*^wt/GA^, *Flnc*^wt/AGA^ × *Flnc*^wt/AGA^ and *Flnc*^wt/AGA^ × *Flnc*^wt/GA^. As a result, a total of 79 *Flnc*^wt/wt^, 98 *Flnc*^wt/GA^, 42 *Flnc*^wt/AGA^ and 18 *Flnc*^AGA/GA^ animals were born (Table 1a,b) and genotyped in F1–F4 generations. No homozygous animals were born, suggesting homozygosity of either allele is embryonically lethal. Surprisingly, compound heterozygous (*Flnc*^AGA/GA^) mice were viable, and the distribution of genotypes in the offspring of the *Flnc*^wt/AGA^ × *Flnc*^wt/GA^ crossing agreed with expectations (Table 1c).

### 2.2. Homozygous Flnc^GA/GA^ Embryos Do Not Survive Beyond E11.5–E12.5

To determine the developmental stage when homozygous embryos die, dated pregnancies from *Flnc*^wt/GA^ × *Flnc*^wt/GA^ crossing were set up. As according to Mendelian’s law of segregation 25% of the offspring should be homozygotes, genotyping of embryos was performed to determine which *Flnc* alleles they carry. For each embryo, a genotype and a microphotograph captured before tissue sample removal were matched. A comparison of the phenotypes of embryos with *Flnc*^GA/GA^, *Flnc*^wt/GA^ and *Flnc*^wt/wt^ genotypes was performed. Morphological features characteristic of each developmental stage, such as the presence and development of various brain regions, eyes, limb buds, and other, were assessed and compared to atlases of normal development [29].

No difference was observed between the wild-type and *Flnc*^wt/GA^ groups; embryos of both genotypes were developing normally. At the same time, *Flnc*^GA/GA^ homozygotes develop at a normal rate only until embryonic day 9.5 (E9.5) (Figure 2). At E10.5 *Flnc*^GA/GA^ embryos lack properly developed forebrain hemispheres and forelimb and hindlimb buds that are characteristic of this developmental stage. At E11.5, the developmental delay of *Flnc*^GA/GA^ embryos is even more pronounced, general embryo tissue degradation is observed, and by E12.5, homozygous embryos are presumably completely resorbed—no *Flnc*^GA/GA^ samples were found at this stage, while empty deciduae were observed.

### 2.3. Mutant mRNA Presence in Muscle Tissue Is Reduced in Heterozygous Mice, but Not in Flnc^AGA/wt^ Hearts

We performed a variant-discerning assessment of mRNA levels in the skeletal muscle and hearts of adult wild-type, *Flnc*^GA/wt^, *Flnc*^AGA/wt^, and *Flnc*^GA/AGA^ mice using qPCR with fluorescent probes.

The data obtained indicate that in skeletal muscle (femoral muscle), mutant variant mRNA is substantially ablated in simple heterozygous mice (*Flnc*^GA/wt^ and *Flnc*^AGA/wt^) (Figure 3A) (*p* = 0.47 and 0.13, respectively, *t*-test). Notably, levels of wild-type mRNA were also slightly, yet insignificantly (*p* > 0.3) decreased compared to its level in wild-type mice.

In the heart, a similar profile was observed, except, strikingly, the mutant variant was present in Flnc^AGA/wt^ samples at a normalized fluorescence comparable to the wild-type probe level (*p* = 0.16 for difference) (Figure 3B). Additionally, the mutant mRNA level seems somewhat decreased in hearts compared to skeletal muscle in compound heterozygous mice, although high variance does not allow statistical assumptions to be made (*p* = 0.47).

Wild-type variant mRNA expression in compound heterozygotes was found to be absent, as expected, in both types of muscle tissue. 

This could suggest that different mechanisms take place to impede mutant protein production in cardiac and skeletal muscles. 

### 2.4. Indicators of Grip Strength and Endurance of Mice with Genotypes Flnc^wt/AGA^, Flnc^wt/GA^, and Flnc^AGA/GA^ Distinguish from Wild-Type Animals

Next, we have studied the influence of mutant FLNC protein expression on the development of pathological phenotype in vivo. Taking into consideration the fact that the FLNC protein is normally expressed and present not only in cardiac Z-discs but also in sarcomeres of striated skeletal muscles, we have hypothesized that introduced mutations may affect the muscle activity of mutant animals. To test the hypothesis, we have performed two physiology tests—endurance and grip strength tests. It is important to note that the assessment of muscle system functions, such as endurance and strength, in homozygous *Flnc*^GA/GA^ and *Flnc*^AGA/AGA^ animals was impossible due to embryonic lethality.

Four groups of 1.5–3 month-old males were used for both tests: *Flnc*^wt/wt^, *Flnc*^wt/GA^, *Flnc*^wt/AGA^, and *Flnc*^AGA/GA^.

Surprisingly, quantitative analysis has revealed a statistically significant increase in grip strength in *Flnc*^AGA/GA^ mice compared to wild-type animals (*p* = 0.0017, Mann–Whitney U-test) (Figure 4). Muscle endurance values were similar in all experimental groups and wild-type animals (*p* = 0.53, Kruskal–Wallis test).

### 2.5. Protein Alterations Do Not Impair Heart Function 

To evaluate the role of mutations in the *Flnc* gene on cardiac function we performed electrocardiogram (ECG) on double mutant mice (*Flnc*^AGA/GA^) and compared it with the ECG of wild-type mice. All mice exhibited a regular sinus rhythm in their ECG data. The mean values (MeanNN) and standard deviations (SDNN) of the R-R intervals did not show significant differences between the two groups (*p*-value MeanNN = 0.49, *p*-value SDNN = 0.69; Mann–Whitney U-test) (Table 2, Figure 5). Thus, heart rate variability was comparable across groups. No specific qualitative differences were observed in the configuration of the atrioventricular complexes between the double mutant mice (*Flnc*^AGA/GA^) and control mice, with amplitudes remaining within normal ranges and exhibiting only individual variations. Comprehensive ECG analysis revealed no abnormalities in excitability or conduction (Figure 6). Consequently, the ECG findings indicated no significant R-R interval abnormalities in the ECG of the double mutant mice (*Flnc*^AGA/GA^).

The comparison of heart rate variability revealed no significant differences between the wild-type mice (WT) and the double mutant mice (*Flnc*^GA/AGA^). The following metrics were analyzed: MeanNN, which is the average of the R-R intervals; SDNN, the standard deviation of the R-R intervals; RMSSD, the square root of the mean of the squared differences between successive R-R intervals; SDSD, the standard deviation of successive differences between R-R intervals; CVNN, the coefficient of variation of R-R intervals, calculated as the standard deviation of the R-R intervals (SDNN) divided by the mean of the R-R intervals (MeanNN); CVSD, the root mean square of successive differences (RMSSD) divided by the mean of the R-R intervals (MeanNN); and MedianNN, the median of the R-R intervals.

Moreover, we performed echocardiography to analyze in detail if there are some differences in the heart function of Flnc^AGA/GA^ mice in comparison to wild-type animals. In both groups of mice, there were no significant pathological findings on echocardiography. Heart chambers were of comparable size in both groups (*p* = 0.8, Mann–Whitney U-test), no zones of reduced activity were detected, and the myocardium without significant differences in thickness between groups in systole (*p* > 0.9, Mann–Whitney U-test) and diastole (*p* = 0.6, Mann–Whitney U-test). No significant differences in fractional shortening (*p* = 0.6, Mann–Whitney U-test) and ejection fraction (*p* = 0.7, Mann–Whitney U-test) indicate functional stability of the myocardium in the animals studied. The data demonstrated comparable systolic and diastolic myocardial function in control and double mutant mice (*Flnc*^AGA/GA^) (Figure 7, Table 3).

### 2.6. Myocardium and Skeletal Muscle Histological Sections of Flnc^wt/GA^, Flnc^wt/AGA^ and Flnc^AGA/GA^ Mice Do Not Exhibit Pathological Features Characteristic of Myopathy

We have prepared histological sections of the cardiac and skeletal muscle of adult *Flnc*^wt/GA^, *Flnc*^wt/AGA^ and *Flnc*^AGA/GA^ animals. Representative images of heart and skeletal muscle sections from animals of different genotypes stained with H&E (hematoxylin and eosin) and HBFP (haematoxylin-basic fuchsin-picric acid) are shown in Figure 8.

Sections show normal cross-striated structure, absence of obvious cellular aggregates and fibrosis. Cardiomyocyte nuclei have normal peripheral cellular localization. The HBFP method is used to visualize fibrin and connective tissue structures in ischemia-damaged myocardium—fuchsin stains damaged regions red, while intact myocardial tissue is yellow to yellow-brown, allowing thorough evaluation of cardiac muscle condition. Using HBFP staining, we have found that heterozygous *Flnc*^wt/GA^ animals do not show any difference compared to wild types, and their cardiac tissue is unaffected.

### 2.7. Computational Prediction of Mutant Proteins Structure

We have performed a computational analysis of the possible effects of introduced mutations on the structure and function of the FLNC protein. Using AlphaFold2 software (ColabFold v1.5.5: AlphaFold2 using MMseqs2), a 3D-model of the spatial arrangement of native filamin C domain 22 loops was built (Figure 9A). Based on the previously solved and published domain 14–15 structure (PDB ID: 7OUU), as well as the known filamin A structure (PDB ID: 6D8C), it can be assumed that loops 1, 3 and 5 are located at the region of interaction between domains 22 and 23. This interaction might be due to two hydrophobic surfaces being in close proximity or the formation of hydrogen bonds between amino acids of closely located alpha helices.

The delGA variant leads to a frameshift and translation termination six amino acid residues downstream of the mutation site, potentially affecting Ig-like domain 22 folding. The homodimerization domain is missing in this variant due to being downstream of the premature stop codon.

The variant delAGA does not lead to a frameshift, but it results in a residue substitution p.E2472D and deletion of Asn2473. Further protein synthesis is not affected in this case. 

Both mutations cause structural changes in loop 5 (Figure 9B). The variant delAGA leads to a loss of a single hydrophilic amino acid, possibly resulting in a reduced number of hydrogen bonds compared to the native protein. As the exact position of the Asn2473 side chain is unclear, its deletion may reduce the interaction between loop 5 and loop 1 or loop 3 or weaken the interaction between domains 22 and 23. In both cases, the weakening of the interaction leads to an insufficiently strong binding, potentially making it difficult to form a functional homodimer.

The *Flnc*^GA^ mutation does not cause a reduction in the number of hydrophilic amino acids in loop 5, although loop 6 and domains 23–24 disappear from the protein due to the frameshift (Figure 9C).

## 3. Discussion

In humans, heterozygous mutations in the *FLNC* gene can cause various types of cardiomyopathies: arrhythmogenic (ACM), dilated (DCM) and restrictive (RCM) [30,31,32]. However, modelling this pathology in mouse strains is challenging as homozygous animals are either embryonically lethal [26,33], or die soon after birth at P0 [34], whereas heterozygous animals do not have any pathological phenotype [34]. 

The phenotype of animal strains reported in this study is in agreement with previously published data—both mutations are embryonically lethal. *Flnc*^GA/GA^ embryos start displaying developmental delay between E9.5 and E10.5, while heterozygous *Flnc*^wt/GA^ embryos develop normally. At E9.5, *Flnc*^GA/GA^ embryos are morphologically indistinguishable from controls, suggesting that the earliest stages of cardiogenesis, starting from cardiogenic mesoderm specification up to at least heart tube rotation, are unaffected. At later stages, the mutation in the *Flnc* gene starts to interfere with normal development and a pathological phenotype develops—at E10.5, homozygous *Flnc*^GA/GA^ embryos display developmental delay, and by E12.5, homozygous embryos are not found, suggestive of their death and absorption. Most probably, the reason behind *Flnc*^GA/GA^ embryo lethality is cardiac dysfunction, as the timing of developmental delay onset is consistent with the start of heart functioning in normally developing mouse embryos [35]. Similarly to the *Flnc*^GA^ mutant strain, analysis of offspring genotypes in the *Flnc*^wt/AGA^ × *Flnc*^wt/AGA^ crossing revealed an absence of homozygous *Flnc*^AGA/AGA^ animals, suggesting embryonic lethality. Moreover, the observed heterozygous to wild-type genotypes ratio is 1:1, while the expected ratio would be 2:1, suggesting that *Flnc*^wt/AGA^ are also prone to embryonic death.

Both physiology tests and histology analyses of skeletal and cardiac muscles performed on heterozygous *Flnc*^wt/GA^ and *Flnc*^wt/AGA^ animals did not reveal any difference as compared to wild-type mice. This suggests that the presence of a single normal allele of *Flnc* is sufficient for normal muscle functioning, which is supported by wild-type mRNA expression evidence. Yet it remains unclear whether the presence of mutant mRNA in the hearts of *Flnc*^wt/AGA^ contributes to the phenotype and at what level the expression regulation takes place: pre- or post-transcriptionally, in all genotype variants.

Surprisingly, compound heterozygous animals (*Flnc*^GA/AGA^) developed normally, were born alive, reached puberty, and displayed statistically significantly higher grip strength as compared to wild-type animals. The DNA samples of these *Flnc*^GA/AGA^ mice were re-sequenced to confirm the compound heterozygous genotype and the absence of the wild-type allele. ECG and Eco-CG have revealed the absence of any difference between compound heterozygous and wild-type mice. 

Based on the data on the relative positions of native protein domains during dimerization [13] we proposed a hypothetical model of “heterodimerization” of two mutant proteins, where mutual compensation of their “weak points” occurs. The FLNCdelGA variant lacks domains 23–24, required for dimerization, while homodimerization of the FLNCdelAGA variant is not strong enough for functional dimer formation. The possible interaction between FLNCdelAGA and FLNCdelGA in the case of heterodimerization in relaxed and tense states is shown in Figure 10.

Normally, domain 24 is responsible for dimerization, while in a protein molecule synthesized from the delGA allele domains 23 and 24 are missing.

Such domain position may explain the fact that heterozygous delGA mice perform better in physical tests—instead of a single domain, three domains are involved in dimerization in the truncated protein in the proposed model.

In the wild-type FLNC protein, domain 24 is responsible for dimerization. At the same time, as suggested by Zhenfeng Mao and Fumihiko Nakamura [13], when relaxation and tension of the muscle occurs, the remaining domains of the FLNC protein are rearranged. When the muscles are tense, the protein stretches into a line in which all domains are located one after the other, and in a relaxed state, the domains form a more specific structure. Based on their hypothesis, we assume that the interaction of two FLNC proteins with different frameshifts results in the interaction of different domains and mutual compensation of their weaknesses. At the same time, in a relaxed state, domain 24 of the FLNC delAGA variant interacts with three domains of the FLNC delGA variant: with domains 18, 20 and the mutant 22 domain. This can lead to increased connectivity in the FLNC dimer and, therefore, mice with that genotype show better strength in comparison to WT animals. In the tense state, the mutant domain 22 of the FLNC delGA variant probably interacts with domain 24 of FLNC delAGA. Thus, according to our proposed model, the main reason for the survival of mice with two variants of FLNC may be due to the changes in the structure of the FLNC dimer in a relaxed state.

Filamin C has not only a structural role in muscle cells but also serves as a mechano-signaling molecule. Thus, it has recently been shown that the key region for mechano-sensing in FLNC is d20 (20th domain of 24 immunoglobulin-like, Ig-like, domains–21-d24) [36]. There is a FILIP1 (filamin-A-interacting protein 1)-mediated molecular mechanism associated with the 20th domain which allows the performance of damaged FLNC degradation. It can be suggested that due to the changes in folding of the mutated variants of FLNC, the mechanosensitive molecular mechanism of their degradation can also be impaired resulting in differences in FLNC stability in comparison to the WT variant. However, this issue requires further detailed study.

Mouse models may be useful for finding specific regions of the Flnc gene which are crucial for normal muscle, particularly heart, development. Thus, in the current work and in the work of Zhou et al. [37], it has been shown that homozygous Flnc mutants are embryonically lethal. Apparently, embryonic death occurs due to impaired heart function. Moreover, animal models may be useful for the investigation of FLNC protein localization and their counteraction with other proteins.

Based on the findings of the current work, it can be concluded that for the restoration of muscle function, not only the WT allele is required, but in some cases, the presence of another mutant allele may be effective. Therefore, it seems a promising perspective to develop of gene therapeutic agents which will introduce additional copies of the FLNC gene or solely some parts of this gene into the cells. Delivery of the whole gene into the cells may be difficult due to the limited capacity of delivery systems (such as viruses or liposomes). However, it can be suggested that restoration of mutated FLNC properties can also be achieved if a truncated protein corresponding to some domains of the FLNC gene is synthesized in cells. Thus, the development of such gene therapeutical medicines may be considered in the future.

Apparently, this study serves to illustrate that mice are not always a good model of human pathological conditions. The reasons behind this discrepancy between mice and humans could be lifespan difference or physiological features, such as circulating blood volume.


**
Summary
**


Using the CRISPR/Cas9 technique we generated two mouse strains carrying filamin C mutations in the 20th domain. One of those mutations (AGA) resulted in an E>>D substitution and N deletion. The other strain (GA) carries a deletion of GA nucleotides leading to a frameshift and a premature stop codon. Homozygous animals of both strains (*Flnc*^AGA/AGA^ and *Flnc*^GA/GA^) were embryonically lethal; *Flnc*^GA/GA^ embryos died prior to the E12.5 stage. The most likely reason for this is impairment in heart function. Heart tissue and skeletal muscles of heterozygous strains carrying mutations in different combinations (*Flnc*^GA/wt^, *Flnc*^AGA/wt^ and *Flnc*^GA/AGA^) did not differ from those obtained from WT animals. Heterozygous animals of both strains (*Flnc*^GA/wt^, *Flnc*^AGA/wt^) are functionally indistinguishable from wild-type animals when performing grip strength and endurance tests. Unexpectedly, compound heterozygous mice (*Flnc*^GA/AGA^) are viable, develop normally, reach puberty, and their cardiac muscle is functionally normal (verified by ECG and Eco-CG). Moreover, *Flnc*^GA/AGA^ mice demonstrated better results in the grip strength physiological test in comparison to WT animals. We also propose a structural model that explains the complementary interaction of two mutant variants of filamin C. The main reason for the survival of *Flnc*^GA/AGA^ mice may be due to the changes in the structure of the FLNC dimer in a relaxed state.

## 4. Materials and Methods

### 4.1. Animals

#### 4.1.1. Animal Housing

Females of the CD1 strain were used as surrogate mothers for genetically edited embryos, and F1 CBA × C57Bl/6J hybrids were used as embryo donors. To obtain pseudopregnant surrogate mothers, females were mated with CD1 strain vasectomized males. C57Bl/6J males were used as breeders. The animals were given ad libitum access to water and specialized feed. The dark-light cycle was 14/10, and the air temperature in the facility was 23 ± 1 °C.

All experiments involving animals were carried out in adherence to local regulations and approved by the IGB RAS bioethics committee.

#### 4.1.2. Transgenic Animals’ Generation

Transgenic mice were obtained by a standard protocol as described previously [38]. Briefly, after breeding prepubertal females (3 weeks old) with mature males (2–3 months), zygotes were obtained. Superovulation was induced to increase the number of received zygotes. Two hormonal injections were used to induce superovulation: 5ME of FSH followed by the injection of 5ME hCG 48 h later. The genetic construct was injected into the cytoplasm of zygotes, which were cultured after microinjection for 1 day in vitro and then embryos that had cleaved into 2 blastomeres were transferred into the oviducts of pseudopregnant females. For embryo transfer, we used pseudopregnant females (CD1 mice, 2–3 months old) that had a vaginal plug on the day when the embryos were transferred. 

On day 19 after embryo transfer, cesarean section and extraction of surviving embryos was performed if natural labor did not occur. In case of natural delivery pups were left with their biological mother for rearing. In case of absence of signs of labor pregnant mice were sacrificed by cervical dislocation, cesarean section was performed and newborn pups were placed with a female who was rearing a litter of a similar age.

### 4.2. Genetic Construct Production

The sgRNA prediction was performed using the online service CRISPOR TEFOR (http://crispor.tefor.net/, accessed on 12 September 2022), considering the subsequent use of SpCas9 nuclease. An efficient sgRNA that cleaves the intended site of mutation was selected.

The sgRNA was cloned into a px330 vector (Addgene; https://www.addgene.org/42230/, accessed on 13 November 2024) as per the manufacturer’s protocol for plasmid microinjection. The vector was cleaved using BbsI-HF restriction enzyme (New England Biolabs, Ipswich, MA, USA) and dephosphorylated with FastAP enzyme (Thermo Fisher Scientific, Waltham, MA, USA), followed by agarose gel electrophoresis for product separation and extraction using the Monarch DNA Gel Extraction Kit (New England Biolabs, Ipswich, MA, USA). Oligonucleotides were synthesized by Evrogen company (Moscow, Russia) as follows: 5′-CACCATTGAGTGGACACCGTTCTCA-3′, 5′-ATTGAGTGACACCGTTCTCAGTTT-3′. Oligonucleotides were phosphorylated by polynucleotide kinase (Thermo Fisher Scientific, USA), annealed and ligated with previously prepared linearized vector using T4 DNA ligase (Thermo Fisher Scientific, USA). The ligation mixture was transformed into competent XL1blue strain cells (Evrogen, Moscow, Russia). Colony screening was performed using a PCR kit (Isogen Lifescience B.V., Utrecht, The Netherlands) and U6-forv primer (GAGGGCCTATTTCCCATGATT) and a reverse primer used for cloning. Two sgRNA-containing colonies were cultured overnight and plasmid DNA was extracted using the Monarch Plasmid Miniprep Kit (New England Biolabs, Ipswich, MA, USA). Plasmid sequences were analyzed by Sanger sequencing; analysis was performed by Evrogen company using the U6-forv primer. Sequencing results were analyzed with Chromas 2.5.0 software (Chromas Lite).

The microinjection mix contained 1 ng/µL px330 vector with sgRNA insertion.

### 4.3. Genotyping

#### 4.3.1. F0 Genotyping

F0 genotyping was carried out by sequencing genomic DNA. The region of interest was amplified using *Flnc* seq F (GTGCCTGACCTACCAAGGCA) and *Flnc* seq R (CATCTTTCGCGGTGCAGCTT) primers, and the obtained fragment was sequenced using the *Flnc* seq R primer.

Genotyping of the F1 and further generations was carried out using real-time PCR with fluorescent probes.

#### 4.3.2. F1 and Further Generations Genotyping

Mice and embryos were genotyped using real-time PCR. Ear samples were taken from 1- to 4-week-old animals for genotyping purposes. DNA was extracted using an alkaline lysis method. Tissue samples were placed in 200 µL of lysis buffer (25 mM NaOH, 0.2 mM EDTA, pH 12) and incubated at 95 °C for 1.5 h, followed by removal of undissolved tissues by centrifugation for 30 s (Microspin FV-2400, bioSan, Riga, Latvia). The DNA-containing supernatant was used as a PCR template, and nucleic acid concentration was measured spectrophotometrically (IMPLEN NanoPhotometer P300, IMPLEN, München, Germany). DNA samples were diluted approximately 100-fold with milli-Q water to reach an optimal concentration of 10 ng/µL DNA per reaction. The diluted DNA sample (3 µL) was added per well in a 96-well plate (SSI Bio, Lodi, CA, USA). In the case of embryo genotyping, 50 µL of lysis buffer was enough to lyse the tissue sample taken from the embryo, and a 10-fold dilution was sufficient. The following primers were used for real-time PCR: forward 5′-GAAGAAGCATGGAGGCCCAC-3′ (Flnc rtgt F), reverse 5′-GCTCCCCAACACGGATCTTG-3′ (Flnc rtgt R) and fluorescent probes FAM-CCCATGAGAACGGTGTCCAC-bhq1 and ROX-CCCATGACGGTGTCCACTCA-bhq2 (for delAGA allele detection), or ROX-CCCATGAACGGTGTCCACTCA-bhq2 (for delGA allele detection).

The delAGA-detecting probe was used in combination with the wild-type-detecting probe to genotype the *Flnc*^GA/AGA^ litter, since no FAM signal was observed in samples of these mice in the presence of the FAM-labeled probe, yet ROX fluorescence was discernible in intensity in simple heterozygous samples.

### 4.4. Expression Analysis (qPCR)

To assess mRNA expression of wild-type and mutant variants, real-time quantitative PCR was used. Each group contained two adult (2–3 months old) animals. Mice were sacrificed by cervical dislocation. A piece of femoral muscle (around 50 mg) and the whole heart were taken. The hearts were incised and quickly rinsed with 0.9% NaCl solution to expel the remaining blood. The tissues were homogenized with a pestle and mortar under liquid nitrogen. Total RNA from tissue powders was extracted using the ExtractRNA kit (Evrogen, Russia). Reverse transcription (1 µg of total RNA per reaction) was performed using the MMLV Revertase kit (Evrogen, Russia) and an oligo-dT(15) primer. 

Quantitative PCR was set up in 96-well plates (SSIbio, Lodi, CA, USA) using SYBR Green as the detection method for *Hprt* reference gene expression, and fluorescent probes (150 nM each) for *Flnc* mRNA variant detection, the same used for genotyping. The primers “Flnc seq F” and “Flnc rtgt R” (Section 4.3) were used to assess *Flnc* expression and amplification specificity using SYBR Green. The delAGA-detecting probe was used to assess mutant mRNA level. A quantity of cDNA equivalent to 30 ng of input RNA was used per reaction. PCR was set up with standard HS-Taq polymerase and run following the protocol: 95 °C for 5 min—initial denaturation, followed by 40 cycles of 95 °C, 30 s—denaturation, 60 °C, 20 s—annealing, 71 °C, 4 s—elongation, and a melt curve analysis. 

Endpoint fluorescence (EF) of probe-containing reactions and threshold cycles (Ct) of SYBR Green reactions were averaged between technical replicates. The obtained mean EFs were normalized to corresponding *Hprt* expressions, calculated as 2 to the power of negative Ct. Prior to this, EFs of the mutant probe were divided by the median ratio of mutant to wild-type EFs (0.504), obtained during genotyping of simple heterozygous mice (*Flnc*^AGA/wt^) to exclude fluorescence efficiency impact and make quantitative comparison possible.

### 4.5. Physiology Test Performing

Tests were carried out on two groups of 1.5- to 3-month-old males, 8–9 animals in each: wild-type controls and an experimental group of heterozygous animals.

#### 4.5.1. Endurance Test

A metal wire (2 mm thick, 50 cm long) was used in the test. The wire was fixed 30 cm above a cage filled with soft bedding to prevent mice from hurting their limbs in case of falling (Appendix A). All mice were weighed prior to the experiment. Mice were grabbed by the tail and hung on the wire with their forelimbs, not allowed to use their hind limbs. The duration of mouse hanging was assessed. The maximal hanging time was limited to 300 s, and if the mouse did not fall within this time, the experiment with the animal was terminated for the day. If the mouse fell after less than 300 s, the duration of hanging was recorded, and the mouse was hung again; 10 attempts in total were given, after which the experiment with the animal was terminated for the day. The test was performed three times with each male from all experimental groups with one-day breaks to let the mice rest (at least 1 min) between independent repeats.

As a result, we obtained data on maximal hanging duration for each animal in triplicates and on the number of attempts, which were analyzed statistically.

#### 4.5.2. Grip Strength Test

A highly sensitive Grip Strength Test Meter for Mice and Rats (“IITC Life Science”, Los Angeles, CA, USA) was used to conduct a grip strength assessment. Animals were grabbed by the tail and brought close to the grabbing metal device, fixed on a sensor (Appendix A). The mouse was allowed to grab the grid only with its forelimbs, after which the animal was gently and slowly pulled backward until it released the grid. The sensor recorded the maximal grip strength in grams. Each animal was given three to five attempts with at least one minute of rest time in between. The test was performed three times with at least one day of rest. The obtained results were normalized to animal body weight measured prior to each test and were analyzed statistically.

### 4.6. Heart Function

#### 4.6.1. Electrocardiography

The functional state of the myocardium in double mutant mice (*Flnc*^AGA/GA^) was evaluated using standard ECG under isoflurane anesthesia [39]. R-R intervals and their variability were analyzed, alongside a qualitative comparison of the atrioventricular complexes between two groups: four double mutant mice (*Flnc*^GA/AGA^) and four control mice (*Flnc*^wt/wt^), all approximately one month old.

ECG recordings were performed at a frequency of 10,000 Hz using the PowerGraph 3.x application. A 5-min recording at a sampling rate of 5000 Hz was analyzed using a custom application developed with Python 3.11 libraries, including neurokit, pandas, numpy, scipy, seaborn, and others. R-R interval data outside the interquartile range (IQR) were filtered. Heart rate variability, defined as the variability of R-R intervals between consecutive heartbeats, was calculated using key metrics [40].

#### 4.6.2. Echocardiography

The cardiac morphofunctional state was evaluated using a Vetus 9 (Mindray, Shenzhen, China) equipped with an L13-3Ns linear probe operated at 13 MHz. Three control and three double mutant mice (*Flnc*^AGA/GA^) were imaged under isoflurane anesthesia. The heart was imaged in 2D and M-mode in the parasternal long- and short-axis views. Parameters of systolic and diastolic ventricular diameter and wall thickness, left ventricular fraction shortening (%), and left ventricular ejection fraction (%) were measured [41].

### 4.7. Tissue Preparation and Histology

For histological study, 4 groups of mice were involved, 5–10 individuals of each genotype: *Flnc*^wt/wt^, *Flnc*^wt/GA^, *Flnc*^wt/AGA^, *Flnc*^AGA/GA^. After intraperitoneal sedation using Vezotil (VETSTEM^®^ pharma & cell, Moscow, Russia) and Xyla (Interchemie werken “De Adelaar” B.V., Waalre, The Netherlands), transcardiac perfusion with saline followed by 10% neutral buffered formalin (NBF) was performed [42]. For the sedation, the mix containing 0.6 mL of Vezotil, 0.3 mL of Xyla, and 9 mL of saline (0.9% NaCl) was used. Each mouse received 0.2 mL of that mix for sedation. Samples of limb muscles and hearts were collected and placed into NBF for 24 h. After the completion of fixation, samples were processed via isopropanol dehydrating solution and mineral oil to paraffin wax according to the manufacturer’s instructions (Medix, Taganrog, Russia). Tissue sectioning was conducted via a rotary microtome RMD-3000 (Medtehnikapoint, Saint Petersburg, Russia). Paraffin sections (5 µm) were cut both for H&E staining and HBPF staining. For H&E staining, Mayer’s Hematoxylin and Eosin aqueous solution (1%) were used (BioVitrum, Saint Petersburg, Russia). HBFP staining (BioVitrum, Russia) was used for early detection of myocardial damage and necrosis. Images were captured on Nikon ECLIPSE Ti (“Nikon”, Tokyo, Japan).

### 4.8. Embryo Dissections for Macroscopic and Histology Analysis

Genetically edited female mice were mated with males carrying the same mutation until the discovery of copulation plug, thus obtaining dated pregnancies. At E8.0–E12.5, pregnant females were sacrificed by cervical dislocation and embryos were collected in saline. After decidua tissue and extraembryonic membrane removal embryo images were captured under a NikonSMZ800N stereomicroscope (Tokyo, Japan), Basler camera (Basler AG, Ahrensburg, Germany) connected via Nikon LT-TV adapter (Japan) using Basler microscopy software (version: 2.1). Next, a tail region of the embryo posterior to hind limb buds was removed and used for DNA extraction for genotyping purposes. The remaining embryo was fixed for histology studies.

### 4.9. Statistical Analysis

Statistical analysis of the data was performed in GraphPad Prism (GraphPad Software 8.0). The average was calculated from independent technical repeats (n of repeats ≥ 3). Statistical significance was assessed using non-parametric analysis, as normal distribution could not be confirmed (according to the Kolmogorov–Smirnov criterion). To detect the difference between two groups, the Mann–Whitney U-test and the Bonferroni correction were applied. The result was considered significant at *p* < 0.008.

For qPCR data comparison, the calculated normalized relative fluorescence values were Log-scaled. Holm–Sidak correction was used for multiple comparisons. 

Bioinformatics analysis was conducted using AlphaFold2 Colab [43]; alignment was performed utilizing Jalview (version 2.11.1.3+dfsg2-5) [44], using the Mafft algorithm with defaults.

## 5. Conclusions

In humans, heterozygous mutations of the *FLNC* gene cause cardiomyopathy [45], and animal models of hereditary cardiomyopathy are required to study possible therapeutic agents and approaches. In this work, we studied two strains of genetically edited mice that carry mutations in the *Flnc* gene to assess if they can be used as a hereditary cardiomyopathy model. This is the first work dedicated to obtaining *Flnc* mutated mice, which contains mutations in the 20th domain of the protein. This domain is not responsible for dimerization of the FLNC protein, although different mutations in this domain lead to embryonic lethality in homozygous state (homozygous *Flnc*^GA/GA^ embryos die at E12.5). Nevertheless, heterozygous animals develop successfully and do not display any cardiomyopathy or skeletal muscle dysfunction signs in adulthood. Surprisingly, compound heterozygosity (*Flnc*^AGA/GA^ genotype) is beneficial compared to homozygosity of either mutant allele. Together, our data suggest that even partial recovery of *Flnc* allele function may restore muscle function, opening up a novel field for development of gene therapy to treat myopathies caused by mutations in this gene. 

## Figures and Tables

**Figure 1 ijms-26-01409-f001:**
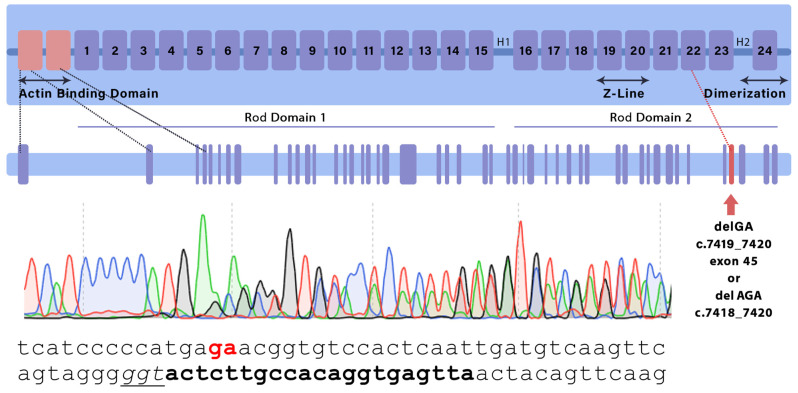
Schematic representation of the FLNC monomer indicating the location of introduced mutations as confirmed by sequencing. Bold and underlined text annotates the sgRNA and PAM sequence, respectively.

**Figure 2 ijms-26-01409-f002:**
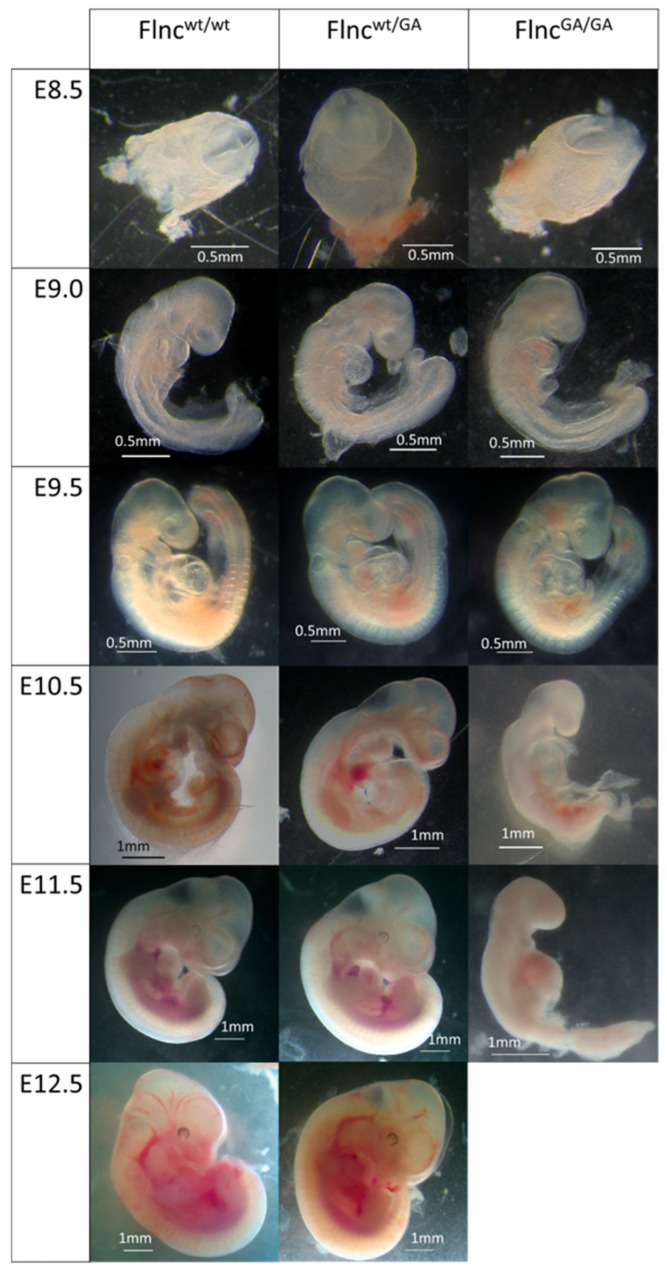
Mouse embryos of different genotypes at E8.5–E12.5 stages; stereomicroscopy.

**Figure 3 ijms-26-01409-f003:**
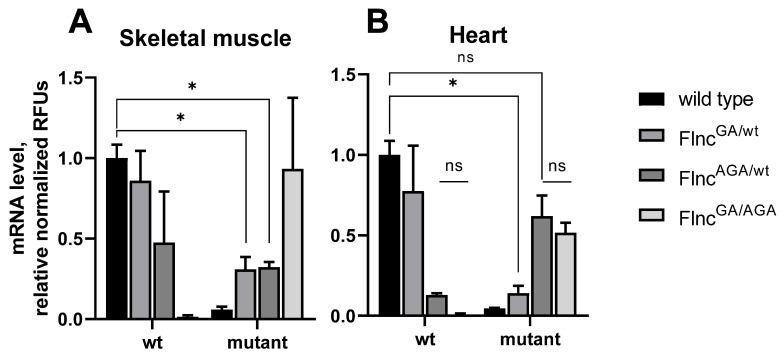
*Flnc* variant (wild-type (wt) or mutant (either delGA or delAGA)) expression evidence at the mRNA level in skeletal muscle (**A**) and heart (**B**). The endpoint fluorescence (relative fluorescence units, RFUs) of probes was measured and normalized to the mean sample *Hprt* expression and to the mean fluorescence of wild-type variant-detecting probes of the wild-type group. The absolute fluorescence of the mutant probe was first normalized to the median ratio of mutant/wt signal of heterozygous *Flnc*^AGA/wt^ mice, obtained during genotyping, to allow comparison between wt and mutant mRNA levels. Data are presented as mean ± SEM. N = 2 for every group. * *p* < 0.05, ns is *p* ≥ 0.05 (Holm-Sidak adjusted).

**Figure 4 ijms-26-01409-f004:**
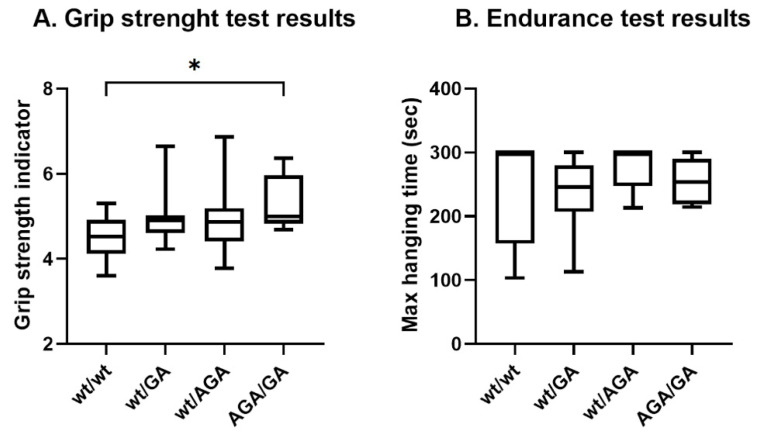
Physiology test results. (**A**) Grip strength test; the grip strength indicator was calculated as an average of grip strength (g) to mouse body weight (g) rates; average values were calculated from three technical repeats; Mann–Whitney U-test, *p* = 0.0161 (*Flnc*^wt/wt^(N = 21)/*Flnc*^wt/GA^(N = 18)), *p* = 0.2226 (*Flnc*^wt/wt^(N = 21)/*Flnc*^wt/AGA^(N = 11)), *p* = 0.0017 (*Flnc*^wt/wt^(N = 21)/*Flnc*^AGA/GA^(N = 9)); (**B**) Endurance test; Mann–Whitney U-test, *p* = 0.6147 (*Flnc*^wt/wt^(N = 9)/*Flnc*^wt/GA^(N = 8)), *p* = 0.5105 (*Flnc*^wt/wt^(N = 9)/*Flnc*^wt/AGA^(N = 5)), *p* = 0.9414 (*Flnc*^wt/wt^(N = 9)/*Flnc*^AGA/GA^(N = 7)). Wiskers on plots refer to min to max, the plot represents the IQR (interquartile range), and the line inside the plot indicates the median; * corresponds to adjusted *p* < 0.008.

**Figure 5 ijms-26-01409-f005:**
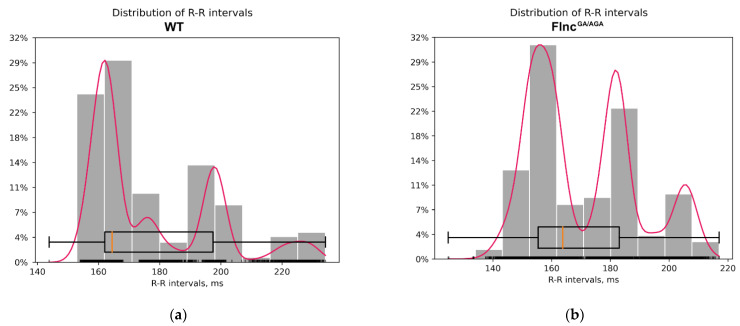
Distribution of R-R intervals: (**a**)—in the wild type mice group (WT); (**b**)—in the double mutant mice group (*Flnc*^GA/AGA^). The red lines outline a kernel density estimation (KDE), the orange lines are placed at distribution medians.

**Figure 6 ijms-26-01409-f006:**
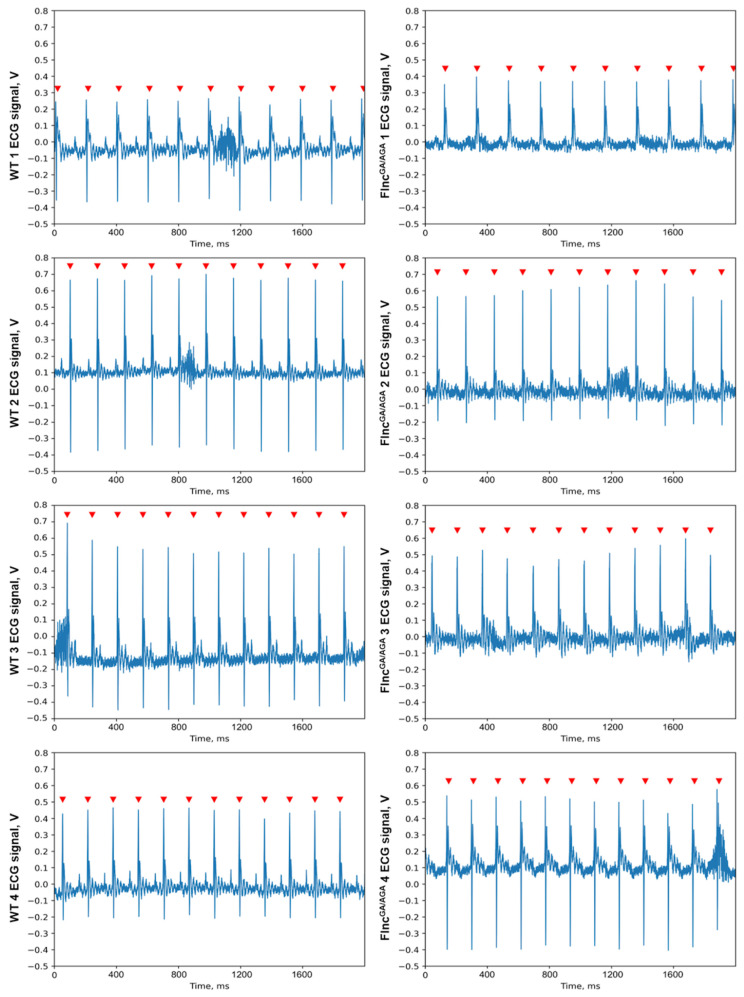
The single-lead ECG segments of wild-type (WT) mice (**left**) and double mutant mice (*Flnc*^GA/AGA^) (**right**). The peaks of the R waves in the ventricular complexes are indicated by red arrows. No specific qualitative differences were observed in the configuration of the atrioventricular complexes between the double mutant mice (*Flnc*^AGA/GA^) and control mice, with amplitudes remaining within normal ranges and exhibiting only individual variations.

**Figure 7 ijms-26-01409-f007:**
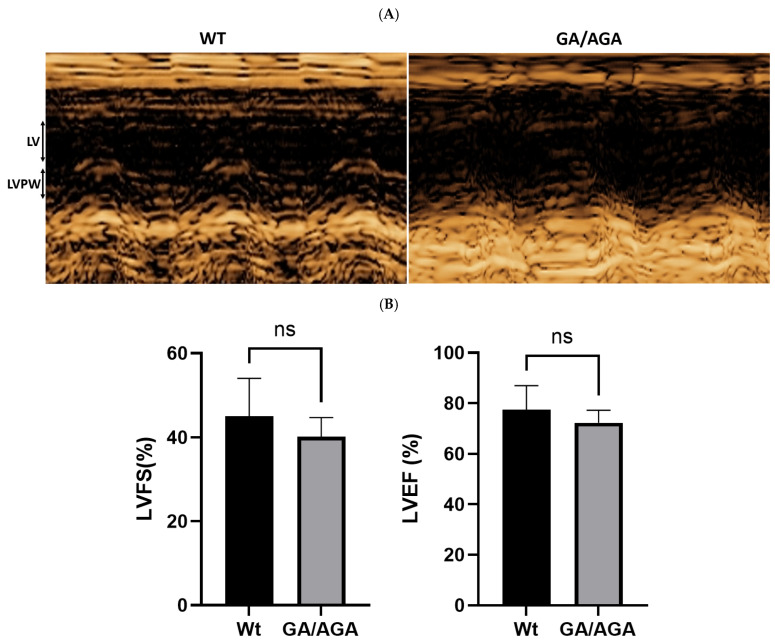
Comparable myocardial function in control and double mutant mice (*Flnc*^AGA/GA^): (**A**)—Echocardiographic M-mode images of left ventricle (LV) and left ventricular posterior wall (LVPW); (**B**)—Echocardiographic analyses of left ventricle fractional shortening (LVFS) and left ventricle ejection fraction (LVEF); ns—indicates statistically non-significant differences.

**Figure 8 ijms-26-01409-f008:**
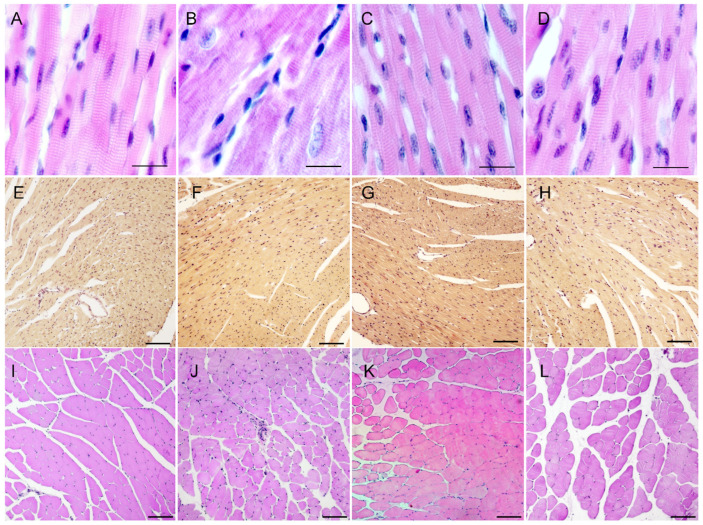
Frontal myocardium sections (**A**–**H**), stained with H&E (**A**–**D**) and HBFP (**E**–**H**), and sagittal sections of gastrocnemius muscles (**I**–**L**), stained with H&E. (**A**,**E**,**I**)—wild-type male; (**B**,**F**,**J**)—*Flnc*^wt/GA^; (**C**,**G**,**K**)—male *Flnc*^wt/AGA^; (**D**,**H**,**L**)—female *Flnc*^AGA/GA^. (**A**–**D**)—scale bar: 20 μm. (**E**–**L**)—scale bar: 100 μm.

**Figure 9 ijms-26-01409-f009:**
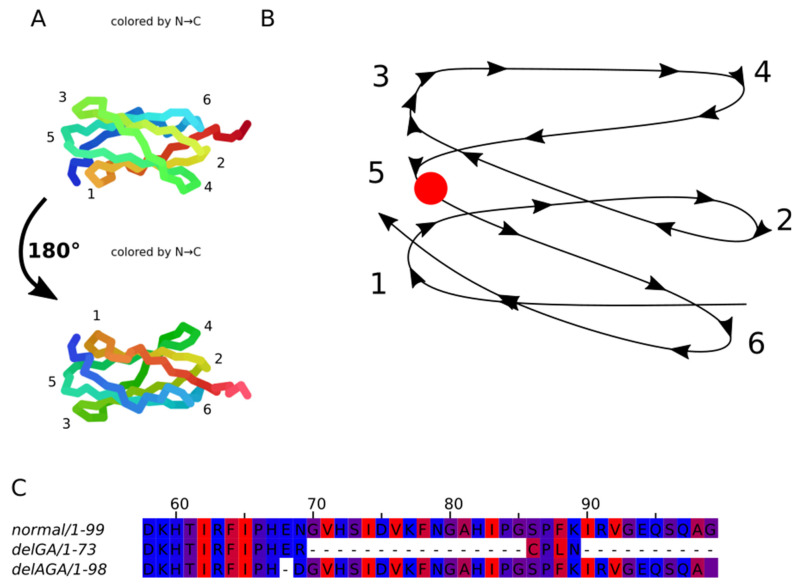
Three-dimensional arrangement of the loops of filamin C domain 22. Structure predicted using AlphaFold2: (**A**)—AlphaFold2 output; (**B**)—schematic loop arrangement. Loops (1–6) are numbered from N- to C-terminus, red dot is for mutation localization; (**C**)—domain 22 sequences alignment from FLNC^wt^ and mutant FLNC^GA^ and FLNC^AGA^. Amino acids are colored according to their hydrophobic/hydrophilic properties: red indicates the most hydrophobic residues, while blue indicates the most hydrophilic ones; the alignment was performed for the domain 22 sequence only.

**Figure 10 ijms-26-01409-f010:**
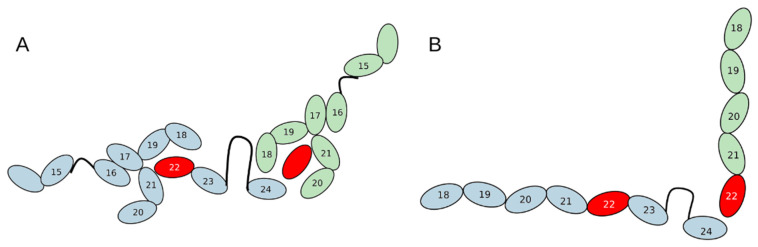
Possible folding of two mutant domains in case of heterodimerization: (**A**)—relaxed state; (**B**)—tense state. The mutant domain is highlighted in red. The delAGA variant is blue, and the delGA variant is green.

**Table 1 ijms-26-01409-t001:** Number of animals of each genotype observed and expected following (**a**) *Flnc*^wt/GA^ × *Flnc*^wt/GA^, (**b**) *Flnc*^wt/AGA^ × *Flnc*^wt/AGA^ and (**c**) *Flnc*^wt/AGA^ × *Flnc*^wt/GA^ crossings.

(**a**) *Flnc*^wt/GA^ × *Flnc*^wt/GA^					
	** *Flnc* ^wt/wt^ **	** *Flnc* ^wt/GA^ **		** *Flnc* ^GA/GA^ **	**Total**
Observed *	32 (28.57%)	80 (71.43%)		0 (0%)	112
Expected	28 (25%)	56 (50%)		28 (25%)	112
(**b**) *Flnc*^wt/AGA^ × *Flnc*^wt/AGA^					
	** *Flnc* ^wt/wt^ **	** *Flnc* ^wt/AGA^ **		** *Flnc* ^AGA/AGA^ **	**Total**
Observed *	29 (51.79%)	27 (48.21%)		0 (0%)	56
Expected	14 (25%)	28 (50%)		14 (25%)	56
(**c**) *Flnc*^wt/AGA^ × *Flnc*^wt/GA^					
	** *Flnc* ^wt/wt^ **	** *Flnc* ^wt/GA^ **	** *Flnc* ^wt/AGA^ **	** *Flnc* ^GA/AGA^ **	**Total**
Observed	18 (26.09%)	18 (26.09%)	15 (21.74%)	18 (26.09%)	69
Expected	17.25 (25%)	17.25 (25%)	17.25 (25%)	17.25 (25%)	69

* denotes *p* < 0.0001 detected by the chi-square test.

**Table 2 ijms-26-01409-t002:** Heart Rate Variability Metrics.

Type of Mice	WT 1	WT 2	WT 3	WT 4	*Flnc*^GA/AGA^ 1	*Flnc*^GA/AGA^ 2	*Flnc*^GA/AGA^ 3	*Flnc*^GA/AGA^ 4
Age	40 days	40 days	30 days	30 days	36 days	36 days	32 days	32 days
Sex	female	male	male	male	male	female	male	male
MeanRR	197.48	196.38	162.54	160.50	197.11	180.67	156.28	155.60
SDNN	4.29	23.04	1.66	2.72	10.23	3.00	7.28	3.34
RMSSD	3.29	2.41	0.90	0.39	2.67	0.74	2.68	1.12
SDSD	3.29	2.42	0.90	0.39	2.67	0.74	2.68	1.12
CVNN	0.02	0.12	0.01	0.02	0.05	0.02	0.05	0.02
CVSD	0.02	0.01	0.01	0.002	0.01	0.004	0.02	0.01
MedianNN	197.8	182.8	162.6	161.6	201.4	181	157.8	155

**Table 3 ijms-26-01409-t003:** Echocardiography metrics.

Type of Mice	WT	GA/AGA
LVPWs, mm	1.4 ± 0.1	1.4 ±0.1
LVPWd, mm	0.56 ± 0.05	0.7 ± 0.17
LVIDs, mm	1.5 ± 0.15	1.8 ± 0.25
LVIDd, mm	2.73 ± 0.4	2.83 ± 0.45
FS, %	45.01 ± 9.02	40.15 ± 4.59
EF, %	77.43 ± 9.56	72.29 ± 4.95

Data are expressed as mean ± SD. LVPWs, left ventricular posterior wall thickness in systole; LVPWd, left ventricular posterior wall thickness in diastole; LVIDs, left ventricle internal diameter in systole; LVIDd, left ventricle internal diameter in diastole; FS, fractional shortening; EF, ejection fraction.

## Data Availability

The data supporting the findings of this study are available within the article and its Appendix A.

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
