# Peer review of "Mutations in Filamin C Associated with Both Alleles Do Not Affect the Functioning of Mice Cardiac Muscles"

_ijms, 2025, doi:10.3390/ijms26041409_

Round 1

Reviewer 1 Report

Comments and Suggestions for Authors

The authors describe the genetic and physiological characteristics of the two created mouse strains with two rode mutations in the Filamin C protein known for its mechanosignaling functions in the Z-disc region of skeletal and cardiac muscles. Both mutations turnede out to be lethal in embryos in homozygotes and did not differ from wildtypes in heterozygotes. The compound animals also did not differ from the wild type mice. The study was well-designed, the data were processed correctly and the conclusions were supported by the results obtained. The results are seemed to be useful for the future development of the genotherapy means. 

However I would like to present two minor cocerns:

1. Line 256 The sentence is incorrect. What does it mean "skeletal and cross-striated muscles"? I guess the authors wanted to write "skeletal and cardiac muscle". That is correct.

2. The authors did not discuss the mechanosignaling functions of the Filamin C and the possible functional alterations in the mutant animals. It will be useful for the readers if the authors include ther paragraph of the possible alterations in the downstream mechanotransduction pathways of Filamin C. Maybe as a speculation.

Anyway the article seems quite well and after minor revisions should be publishes as soon as possible. 

Author Response

The authors describe the genetic and physiological characteristics of the two created mouse strains with two rode mutations in the Filamin C protein known for its mechanosignaling functions in the Z-disc region of skeletal and cardiac muscles. Both mutations turnede out to be lethal in embryos in homozygotes and did not differ from wildtypes in heterozygotes. The compound animals also did not differ from the wild type mice. The study was well-designed, the data were processed correctly and the conclusions were supported by the results obtained. The results are seemed to be useful for the future development of the genotherapy means. 

However I would like to present two minor cocerns:

  1. Line 256 The sentence is incorrect. What does it mean "skeletal and cross-striated muscles"? I guess the authors wanted to write "skeletal and cardiac muscle". That is correct.
  2. The authors did not discuss the mechanosignaling functions of the Filamin C and the possible functional alterations in the mutant animals. It will be useful for the readers if the authors include ther paragraph of the possible alterations in the downstream mechanotransduction pathways of Filamin C. Maybe as a speculation.

Anyway the article seems quite well and after minor revisions should be publishes as soon as possible. 

First of all, we would like to thank the reviewer for the time and efforts spent studying the materials of our article. We tried to take into account all the recommendations as much as possible, and below we present our answers to the questions asked in the format of a dialogue (Q and A). All changes are highlighted in yellow.

Q1. 1. Line 256 The sentence is incorrect. What does it mean "skeletal and cross-striated muscles"? I guess the authors wanted to write "skeletal and cardiac muscle". That is correct.

A1. Thank you very much. We fixed this error. You are right, there should be "skeletal and cardiac muscle".

Q2. The authors did not discuss the mechanosignaling functions of the Filamin C and the possible functional alterations in the mutant animals. It will be useful for the readers if the authors include ther paragraph of the possible alterations in the downstream mechanotransduction pathways of Filamin C. Maybe as a speculation.

A2. We added this at the end of Discussion section.

Reviewer 2 Report

Comments and Suggestions for Authors

The authors created 2 mouse strains carrying filamin C mutations — one of them has a deletion of three nucleotides at position c.7418_7420, The other strain carries a deletion of GA nucleotides at position c.7419_7420. They have shown that heterozygous animals of both strains are indistinguishable from wild type animals in terms of function and morphology, while homozygosity is embryonically lethal. However, compound heterozygous mice are viable with their normal cardiac function. In additional,  they also propose a potential hypothesis from a protein structural prospective.  Those findings might provide insights into cardiomyopathy caused by FLNC mutations. Overall, data are presented in a clear way and readers can easily follow. However, the manuscript might be improved through addressing the following points.

Major points:

1. ECG data shown in Figure 5 was presented with traces only. Further quantifications other than R-R interval can be done to support the claims authors have made.

2. It was not clear explained what a knockout group refer to. Does it refer to  FlncAGA/GA group? Also, confirming the absence of FLNC at a protein level will be necessary.

3. Only the ECG data were used to assess cardiac function. Additional data such echocardiogram will be important for readers to evaluate the conclusion

Minor points:

1. In Figure 1, both mutations were labeled as delGA.

Author Response

The authors created 2 mouse strains carrying filamin C mutations — one of them has a deletion of three nucleotides at position c.7418_7420, The other strain carries a deletion of GA nucleotides at position c.7419_7420. They have shown that heterozygous animals of both strains are indistinguishable from wild type animals in terms of function and morphology, while homozygosity is embryonically lethal. However, compound heterozygous mice are viable with their normal cardiac function. In additional,  they also propose a potential hypothesis from a protein structural prospective.  Those findings might provide insights into cardiomyopathy caused by FLNC mutations. Overall, data are presented in a clear way and readers can easily follow. However, the manuscript might be improved through addressing the following points.

Major points:

  1. ECG data shown in Figure 5 was presented with traces only. Further quantifications other than R-R interval can be done to support the claims authors have made.
  2. It was not clear explained what a knockout group refer to. Does it refer to  FlncAGA/GA group? Also, confirming the absence of FLNC at a protein level will be necessary.
  3. Only the ECG data were used to assess cardiac function. Additional data such echocardiogram will be important for readers to evaluate the conclusion.

Minor points:

  1. In Figure 1, both mutations were labeled as delGA.

First of all, we would like to thank the reviewer for the time and efforts spent studying the materials of our article. We tried to take into account all the recommendations as much as possible, and below we present our answers to the questions asked in the format of a dialogue (Q and A). All changes are highlighted in yellow.

Q1. ECG data shown in Figure 5 was presented with traces only. Further quantifications other than R-R interval can be done to support the claims authors have made.

A1. Thank you for your helpful comment, and we appreciate your attention to detail. Indeed, Figure 5 presents only filtered ECG data. Unfortunately, due to high noise levels in the signal, we were unable to obtain additional parameters beyond the analysis of R-R intervals. We agree that further quantitative data could strengthen our conclusions. However, based on the current data, no specific qualitative differences in heart automaticity, conductivity, or excitability were observed between wild-type and knockout mice.

Q2. It was not clear explained what a knockout group refer to. Does it refer to  FlncAGA/GA group? Also, confirming the absence of FLNC at a protein level will be necessary.

A2. We agree with the reviewer that it will be interesting to measure the levels of different isoforms of FLNC proteins in the murine muscles. Unfortunately, we suppose that we are unable to confirm the absence of FLNC protein isoforms because they should present. Designated mutations (AGA and GA) do not lead to prevention of FLNC protein synthesis. AGA mutation lead to deletion of three nucleotides in exon 45 at position NM_001081185.2:c.7418_7420, and, thus, an amino acid substitution p.E2472D and Asn2473 deletion, and in another one GA deletion at NM_001081185.2:c.7419_7420 led to a frameshift mutation and a premature stop codon. Therefore at least some parts of FLNC proteins should persist in case of both considered mutations. Moreover, it is very difficult to find commercially available antibodies which will be specific for that mutated FLNC isoforms. Production of antibodies which will allow to detect mutated FLNCdelAGA protein (that form of FLNC protein differs from WT protein only by 2 aminoacids) is a separate and very difficult task. Moreover, now we are unable to use commercially available antibodies for FLNC protein for WT protein staining, their shipping to our country may takes at least 2-3 months. We agree with the reviewer that investigation at protein level of FLNC (for example, its localization) is very interesting and in future should be performed, however now we are unable to perform it due to above mentioned reasons.

To clarify which knockout group was used in ECG experiments we added additional sentence in the beginning of the chapter 2.4 “To evaluate the role of mutations in Flnc gene on cardiac function we performed electrocardiogram (ECG) of double mutant mice (FlncAGA/GA) and compared it with ECG of wild-type mice.”

Q3. Only the ECG data were used to assess cardiac function. Additional data such echocardiogram will be important for readers to evaluate the conclusion.

A3. As you recommended, we performed echocardiogram for FlncAGA/GA (double mutant) mice and compared it to WT mice. We did not found any differences between these two groups by this method. We added the chapter 4.52.; expanded the chapter 2.4; and mentioned the absence of differences in discussion section “ECG and Eco-CG have revealed the absence of any difference between compound heterozygous and wild-type mice.”

Q4. In Figure 1, both mutations were labeled as delGA.

A4. Corrected.

Reviewer 3 Report

Comments and Suggestions for Authors

The topic of the article 'Mutations in Filamin C Associated With Both Alleles do not Affect the Functioning of Mice Cardiac Muscles' is highly interesting to a broad readership!

I have only some minor changes, which could help to improve the quality of this nice manuscript:

1.) I would write human genes in Capitals and Italics and would differentiate them from the murine genes (FLNC vs Flnc).

2.) Please add a OMIM identifier for FLNC.

3.) Line 39/40: What is with the intercalated disc in the heart?

4.) Line 236; Please add also 'Mutations in FLNC are Associated with Familial Restrictive Cardiomyopathy' 2016, since this is the first report about FLNC mutations in patients with RCM. Beside DCM and ACM, RCM is the third phenotype which can be observed in humans.

In summary, this manuscript is well writen and should be published after the authors followed the minor points (#1-4). In summary, I suggest a minor revision. Good luck!

Author Response

The topic of the article 'Mutations in Filamin C Associated With Both Alleles do not Affect the Functioning of Mice Cardiac Muscles' is highly interesting to a broad readership!

I have only some minor changes, which could help to improve the quality of this nice manuscript:

1.) I would write human genes in Capitals and Italics and would differentiate them from the murine genes (FLNC vs Flnc).

2.) Please add a OMIM identifier for FLNC.

3.) Line 39/40: What is with the intercalated disc in the heart?

4.) Line 236; Please add also 'Mutations in FLNC are Associated with Familial Restrictive Cardiomyopathy' 2016, since this is the first report about FLNC mutations in patients with RCM. Beside DCM and ACM, RCM is the third phenotype which can be observed in humans.

In summary, this manuscript is well writen and should be published after the authors followed the minor points (#1-4). In summary, I suggest a minor revision. Good luck!

First of all, we would like to thank the reviewer for the time and efforts spent studying the materials of our article. We tried to take into account all the recommendations as much as possible, and below we present our answers to the questions asked in the format of a dialogue (Q and A). All changes are highlighted in yellow.

Q1.) I would write human genes in Capitals and Italics and would differentiate them from the murine genes (FLNC vs Flnc).

A1. We corrected that In the text. There are multiple changings, therefore I do not provide full list of lines with these changes. However, all these changes are highlighted in yellow.

Q2.) Please add a OMIM identifier for FLNC.

A2. We added the reference to OMIM database in the line 50.

Q3.) Line 39/40: What is with the intercalated disc in the heart?

A3. Of course, FLNC also is localized to intercalated discs in case of heart muscle. We added this information in the lines 40-41.

Q4.) Line 236; Please add also 'Mutations in FLNC are Associated with Familial Restrictive Cardiomyopathy' 2016, since this is the first report about FLNC mutations in patients with RCM. Beside DCM and ACM, RCM is the third phenotype which can be observed in humans.

A4. We added this reference in the begging of Discussion section.

Reviewer 4 Report

Comments and Suggestions for Authors

1. Expand the introduction with background on FLNC's role in muscle structure and its implications in human myopathies and cardiomyopathies.

2. Provide detailed descriptions of how the mutations were introduced and assessed, and clarify the methods used to study functional and morphological characteristics.

3. Elaborate on the proposed structural model with visual aids and explain its significance in understanding the complementary interaction of mutant variants.

4. Discuss the potential applications of the mouse models in studying FLNC-related diseases or therapies and suggest areas for further investigation.

5. Provide a concise summary of findings to enhance readability and impact.

Author Response

First of all, we would like to thank the reviewer for the time and efforts spent studying the materials of our article. We tried to take into account all the recommendations as much as possible, and below we present our answers to the questions asked in the format of a dialogue (Q and A). All changes are highlighted in yellow.

Q1. Expand the introduction with background on FLNC's role in muscle structure and its implications in human myopathies and cardiomyopathies.

A1. We expanded the Introduction.

Q2. Provide detailed descriptions of how the mutations were introduced and assessed, and clarify the methods used to study functional and morphological characteristics.

A2. We expanded chapter 4.1.2. Information about genotyping and functional test is given in chapters 4.2-4.2. Information about histological preparation of samples is given in chapter 4.6.

Q3. Elaborate on the proposed structural model with visual aids and explain its significance in understanding the complementary interaction of mutant variants.

A3. We added the following text in the Discussion section.

In the wild-type FLNC protein, domain 24 is responsible for dimerization. At the same time, as suggested by Zhenfeng Mao and Fumihiko Nakamura (https://www.mdpi.com/1422-0067/21/8/2696 ) when relaxation and tension of the muscle occurs, the remaining domains of the FLNC protein are rearranged. When the muscles are tense, the protein stretches into a line in which all domains are located one after the other, and in a relaxed state, the domains form a more specific structure. Based on their hypothesis, we assume that the interaction of two FLNC proteins with different frameshifts results in the interaction of different domains and mutual compensation of their weaknesses. At the same time, in a relaxed state, the domain 24 of the FLNC delAGA variant interacts with three domains of the FLNC delGA variant: with 18, 20 and mutant 22 domains. This can lead to increased connectivity in the FLNC dimer and, therefore, mice with that genotype show better strength in comparison to WT-animals. In the tensed state, the mutant domain 22 of the FLNC delGA variant probably interacts with the domain 24 of FLNC delAGA. Thus, according to our proposed model, the main reason for the survival of mice with two variants of FLNC may be due to the changes in the structure of the FLNC dimer in a relaxed state.

Q4. Discuss the potential applications of the mouse models in studying FLNC-related diseases or therapies and suggest areas for further investigation.

A4. We enlarged the end of Discussion section for this.

Q5. Provide a concise summary of findings to enhance readability and impact.

After Discussion section we added chapter “Summary” where main findings are given.

Reviewer 5 Report

Comments and Suggestions for Authors

This is an interesting and carefully performed article. However, there are some problems that set a question mark on the conclusion of this study.

Main points:   

The conclusion that mice might not always optimal to analyze pathophysiology is not a conclusion from your study but known for many years. You may have added another example but why is this important? But in this case the question is whether this is the first report with this gen. An important but missing control would be whether protein expression in heterozygous is different from WT mice and whether ultrastructure of sarcomers is different. Why can the WT allele not simply have replaced the mutant gene? This could explain the strong phenotype in homozygous mice.

Table 2: The value for WT2 SDNN seems to be wrong. Otherwise it is difficult to understand why RMSSD is normal but SDNN is. There seems to be a large discrepancy in HRV between individual mice, making this test not really helpful.

Abstract: We have shown sounds as you have shown this previously. What is new in this report?

In the abstract we need more information about the aim of the study, the nature of these mutations, and finally, about the findings. What did you really measure to come to your conclusions.

Key word cardiomyopathy sounds funny as these mice do not have such a phenotype.

Author Response

This is an interesting and carefully performed article. However, there are some problems that set a question mark on the conclusion of this study.

Main points:   

The conclusion that mice might not always optimal to analyze pathophysiology is not a conclusion from your study but known for many years. You may have added another example but why is this important? But in this case the question is whether this is the first report with this gen. An important but missing control would be whether protein expression in heterozygous is different from WT mice and whether ultrastructure of sarcomers is different. Why can the WT allele not simply have replaced the mutant gene? This could explain the strong phenotype in homozygous mice.

Table 2: The value for WT2 SDNN seems to be wrong. Otherwise it is difficult to understand why RMSSD is normal but SDNN is. There seems to be a large discrepancy in HRV between individual mice, making this test not really helpful.

Abstract: We have shown sounds as you have shown this previously. What is new in this report?

In the abstract we need more information about the aim of the study, the nature of these mutations, and finally, about the findings. What did you really measure to come to your conclusions.

Key word cardiomyopathy sounds funny as these mice do not have such a phenotype.

First of all, we would like to thank the reviewer for the time and efforts spent studying the materials of our article. We tried to take into account all the recommendations as much as possible, and below we present our answers to the questions asked in the format of a dialogue (Q and A). All changes are highlighted in yellow.

Q1.1. The conclusion that mice might not always optimal to analyze pathophysiology is not a conclusion from your study but known for many years. You may have added another example but why is this important? But in this case the question is whether this is the first report with this gen.

A1.1. We moved sentences dedicated to the discussion that mice may not be suitable model for investigating some human diseases at the end of Discussion section. Moreover, we modified conclusion focusing the attention of the reader in the main findings of the work.

Q1.2. An important but missing control would be whether protein expression in heterozygous is different from WT mice and whether ultrastructure of sarcomers is different.

A1.2. We agree with reviewer that evaluation of FLNC at the protein level is very interesting and important task. It will be interesting to measure levels of different isoforms and investigate their localization. Unfortunately, we are unable to perform such experiments right now. Designated mutations (AGA and GA) do not lead to prevention of FLNC protein synthesis. AGA mutation lead to deletion of three nucleotides in exon 45 at position NM_001081185.2:c.7418_7420, and, thus, an amino acid substitution p.E2472D and Asn2473 deletion, and in another one GA deletion at NM_001081185.2:c.7419_7420 led to a frameshift mutation and a premature stop codon. Therefore at least some parts of FLNC proteins should persist in case of both considered mutations. Moreover, it is very difficult to find commercially available antibodies which will be specific for that mutated FLNC isoforms. Production of antibodies which will allow to detect mutated FLNCdelAGA protein (that form of FLNC protein differs from WT protein only by 2 aminoacids) is a separate and very difficult task. Moreover, now we are unable to use commercially available antibodies for FLNC protein for WT protein staining, their shipping to our country may takes at least 2-3 months. We agree with the reviewer that investigation at protein level of FLNC is very interesting and in future should be performed, however now we are unable to perform it due to above mentioned reasons.

Q1.3. Why can the WT allele not simply have replaced the mutant gene? This could explain the strong phenotype in homozygous mice.

Heterozygous strains FlncGA/wt, FlncAGA/wt exhibited normal development and did not differ by multiple parameters (physiological tests, heart and skeletal muscles histology) from WT animals. Therefore, haploinsufficiency is not typical for Flnc in mice.

Q2. Table 2: The value for WT2 SDNN seems to be wrong. Otherwise it is difficult to understand why RMSSD is normal but SDNN is. There seems to be a large discrepancy in HRV between individual mice, making this test not really helpful.

Thank you for your valuable comment. Indeed, heart rate variability (HRV) assessment is an important tool for analyzing myocardial function, but as you rightly pointed out, the results can show significant individual variations, which requires careful interpretation of the data.

In our study, we used standard and widely accepted HRV metrics such as SDNN (standard deviation of RR intervals) and RMSSD (root mean square of successive RR interval differences) to identify potential differences in myocardial function between wild-type and knockout mice. The values for these parameters are presented in Table 2. We believe the values for SDNN and RMSSD are correct, but if there is an error in the SDNN value for the WT2 group, we will verify the data and make any necessary corrections in the final version of the manuscript.

Regarding the variability in HRV between individual mice, this is indeed a known issue and may reflect natural variations in the cardiovascular system of the animals. In our experiment, we aimed to minimize the influence of these individual differences by performing appropriate statistical analyses, including the Mann-Whitney test to detect differences between groups. However, as you noted, the variability in HRV between individual animals can reduce the sensitivity of the test and complicate the interpretation of the data. We discuss this in the manuscript (lines 155-157), where we state that, despite these variations, no statistically significant differences were found between the wild-type and knockout groups.

Additionally, it is important to note that reduced HRV is associated with various pathological conditions, including heart failure with preserved ejection fraction, making this measure valuable in assessing functional changes in the myocardium. However, as you suggested, in order to draw more precise conclusions from these data, it is crucial to account for individual differences between animals, which we will consider in future studies.

Overall, we agree that minimizing the impact of such variations is important for enhancing the reliability of our conclusions, and we will strive for more stringent control of factors affecting HRV in future research.

We appreciate your thoughtful comments, which will certainly help improve the quality of the manuscript.

Q3. Abstract: We have shown sounds as you have shown this previously. What is new in this report? In the abstract we need more information about the aim of the study, the nature of these mutations, and finally, about the findings. What did you really measure to come to your conclusions.

A3. We significantly rewrote the Abstract.

Q4. Key word cardiomyopathy sounds funny as these mice do not have such a phenotype.

A4. We agree with the reviewer that investigated mice do not exhibit signs of cardiomyopathy, nevertheless mutations in the FLNC gene in humans lead to such abnormalities. Therefore, this keyword may be useful for researchers who is looking information about this theme. However, if it is strongly required we are ready to remove this keyword.

Round 2

Reviewer 2 Report

Comments and Suggestions for Authors

Thanks for addressing my questions. My follow-up questions is shown below.

Q1: Considering the ECG is something fundamental for this paper. High noise levels makes data not applicable for further analysis to extract more info. to assess electrical activity of the heart. Thus, authors might be careful on what can be claimed based on existing data. For example, instead of claiming double mutant did not cause any electrical abnormalities, pls consider claiming double mutant did not cause any R-R interval abnormalities in the ECG.

Q2: If ‘knockout’ is just a typo, authors should double check the manuscript to remove this typo as this it is very confusing to readers. If protein levels can not be measured somehow, author should show if the expression of FLNC is changed at mRNA levels.

Q3: none

Q4: none

Author Response

Thank you very much for your commentaries. Below we will provide answers for the questions in the format of a dialogue (Q and A). All changes in the text of the manuscript are highlighted in blue.

Q1: Considering the ECG is something fundamental for this paper. High noise levels makes data not applicable for further analysis to extract more info. to assess electrical activity of the heart. Thus, authors might be careful on what can be claimed based on existing data. For example, instead of claiming double mutant did not cause any electrical abnormalities, pls consider claiming double mutant did not cause any R-R interval abnormalities in the ECG.

A1. Thank you for your valuable feedback. We appreciate your suggestion to refine our claims regarding the ECG data and the electrical abnormalities observed in the double mutant. We agree that high noise levels can complicate the analysis of the heart's electrical activity.To address this, we would like to highlight that the assessment of myocardial electrophysiology in our study was conducted after applying programmed algorithms to reduce ECG noise. This approach allowed for a more accurate and qualitative analysis of the ECG, including the evaluation of R-R intervals. As a result, we are confident that the data presented are reliable within the scope of this noise reduction process.

In light of your comment, we will revise the manuscript to adjust the phrasing and avoid overstatement. Specifically, we will modify the statement to:  "Consequently, the ECG findings indicated no significant R-R interval abnormalities in the ECG of the double mutant mice (FlncAGA/GA).." reflecting a more precise interpretation of the data.

We hope this revision will clarify the methods used and provide a more accurate representation of our findings. Thank you again for your constructive suggestions

Q2: If ‘knockout’ is just a typo, authors should double check the manuscript to remove this typo as this it is very confusing to readers. If protein levels can not be measured somehow, author should show if the expression of FLNC is changed at mRNA levels.

A2. You are right. The term “knockout” is inappropriate for these mutations. We have made the necessary changes in the text of the entire article.

As it has been required, we performed measurements of Flnc mRNA levels by RT-PCR. Novel data are presented in the chapters 2.3 and 4.4. Moreover, we discuss these novel data in Discussion section.

Reviewer 5 Report

Comments and Suggestions for Authors

I thank the authors for their comments. However, if you agree that the study can be significantly improved by the still missing protein measurements and by more complete HRC analysis it remains unclear for me why the authors do not perform this now. The conclusion must be justafied by the expermimenst and at the moment this is not done. Moreover, HRV is a readoout for the balance of sympathetic and parasympathetic activity not for heart function. Reduced HRV idicates sympathetic drive that then may account for HF. 

Author Response

I thank the authors for their comments. However, if you agree that the study can be significantly improved by the still missing protein measurements and by more complete HRC analysis it remains unclear for me why the authors do not perform this now. The conclusion must be justafied by the expermimenst and at the moment this is not done. Moreover, HRV is a readoout for the balance of sympathetic and parasympathetic activity not for heart function. Reduced HRV idicates sympathetic drive that then may account for HF. 

Thank you very much for your commentaries. All changes in the text of the manuscript are highlighted in blue.

Answer.

Thank you for your valuable comment. You are correct that heart rate variability (HRV) reflects the balance between sympathetic and parasympathetic activity. However, I would like to emphasize that reduced HRV is also closely associated with functional and structural changes in the myocardium, particularly in heart failure. Therefore, changes in HRV can be interpreted as a potential sign of myocardial dysfunction - a result from adaptive and maladaptive changes in the myocardium, rather than solely reflecting autonomic imbalance. Moreover, HRV changes may serve as both a consequence and a predictor of heart failure progression, making it a useful marker for assessing the risk of disease development.

Regarding the quality of our ECG analysis, we would like to highlight that, we did not observe any qualitative abnormalities in myocardial conductivity, excitability, or automaticity. Additionally, the lack of significant changes in R-R intervals could be indicative not only of alterations in autonomic regulation but also of the heart muscle's ability to respond to these autonomic changes, suggesting functional adaptation or compensation mechanisms.

We agree with reviewer that investigation of FLNC proteins levels and their localization in Flnc-mutated mice is very interesting (and important!) task. However, currently we are unable to perform this analysis due to the difficulties in antibodies shipping and preparing (it is very difficult to produce antibodies which would make it possible to distinguish FLNC-AGA mutated protein variant from the WT-protein). Nevertheless, we performed RT-PCR to measure the presence of Flnc mRNAs in skeletal muscles and heart.

Thank you again for your thoughtful feedback, which has helped improve the clarity of our interpretation.